# Observer-Based Active Fault-Tolerant Control of an Asymmetric Twin Wind Turbine

**Mariem Makni** [1,2,*] **, Ihab Haidar** [2]**, Jean-Pierre Barbot** [2,3]**, Franck Plestan** [3]**, Nabih Feki** [1,4] **and Mohamed Slim Abbes** [1]

[1] Laboratory of Mechanics, Modelling and Manufacturing (LA2MP), National Engineering School of Sfax, Sfax 3038, Tunisia; fekinabih@gmail.com (N.F.); slim.abbes@enis.tn (M.S.A.)
[2] Quartz Laboratory, EA-7393, ENSEA, 95014 Cergy-Pontoise, France; ihab.haidar@ensea.fr (I.H.); barbot@ensea.fr (J.-P.B.)
[3] LS2N, UMR CNRS 6004, ECN, 44300 Nantes, France; franck.plestan@ec-nantes.fr
[4] Higher Institute of Applied Science and Technology of Sousse, Sousse University, Sousse 4003, Tunisia
[*] Correspondence: mariem.makni@ensea.fr

**Abstract:** This paper deals with the design of an active fault-tolerant control based on observers for a twin wind turbine, consisting of two wind turbines mounted on the same tower. We consider an asymmetric conditions case, when only one turbine is affected by an inter-turn short circuit fault of a permanent magnet synchronous machine. A diagnosis design is developed which combines the fault estimation method together with an active fault-tolerant control. The main advantage of the proposed method is to detect and correct the considered fault in a short time in order that the twin wind turbine behaves as it does in the healthy case.

**Keywords:** diagnosis; active fault-tolerant control; twin wind turbine; electrical fault; permanent magnet synchronous machine



## 1. Introduction

The huge amounts of electricity consumption and limited energy supply have spearheaded the necessity of renewable energy as solar photovoltaics and wind turbines. In this work, a new concept of a twin wind turbine (TWT), which is composed of two turbines mounted on the same tower, is recalled [1]. It is intended to be used on offshore, where severe conditions lead to the appearance of faults. The topic of fault diagnosis of wind turbines during operation has been addressed increasingly in recent years. In fact, because of the non-stationary operating conditions and the variable load, wind turbines are subject to many types of faults, such as blade faults, mechanical faults [2] and electrical faults. Here we consider an electrical fault which affect only one turbine of the TWT. The inter-turn short circuit is the most common fault in permanent magnet synchronous machines (PMSM). The latter is usually related to insulation degradation, that grows and leads to devastating consequences in a short period of time, even if an early detection and diagnosis is not established. The considered fault, which influences only one turbine in our case, leads to consider the problem of control of the TWT under asymmetric conditions. For that, an active fault-tolerant control based on observers is developed. The principle of the method is analyzed at two levels: the existence of fault and its severity are determined (level 1), and the specific fault is then compensated to stabilize the system around a reference trajectory (level 2). Various works have picked up on the real-time detection of this type of failure in the literature [3,4].

A motor current signature analysis (MCSA) technique for inter-turn stator fault detection have been reported in [5]. However, the MCSA is usually used on stationary analysis, which leads to unsatisfactory results when they are applied under non-stationary conditions [6]. To analyze non-stationary conditions, the discrete wavelet transforms (DWT) and

the continuous wavelet transforms (CWT) are used in [7,8] to prove the time–frequency features in the detection of a stator winding fault and a rotor imbalance fault in the wind turbine. The DWT method has a powerful filtering capability for wind turbine signals. The application of the DWT strengthens the viability of the proposed technique in detecting the changes of wind turbine running condition. Indeed, the feasibility of detecting a wind turbine mechanical fault through analyzing the generator power signal using the CWT technique has been demonstrated. The wavelet analysis is capable of detecting the stator turns fault under both stationary and non-stationary conditions. Nevertheless, it is still not able to locate the fault position and to predict the fault severity.

In [9], a diagnosis method based on fuzzy logic is applied to detect inter-turn short circuit and open phase faults in the PMSM of a wind turbine.

Apart from analyzing conventional vibration studies that are articulated on current, voltage, torque and power signals [10], model-based methods, such as parity space techniques, parameters estimation and observers methods [11] represent some of the most successful fault diagnosis approaches. In fact, observers constitute an important tool which can estimate outputs of the system even under faulty condition case. Specifically, sliding mode observers which have been vastly used in many dynamic systems, such as robotics, vehicles and motors. This method allows the identification of the fault and, furthermore, its isolation. This foremost step is called a fault detection and isolation (FDI). For a linear system, two sliding mode observers in cascade are considered in [12]. Observers-based approaches, based on sliding mode techniques, are also reviewed in [13,14] to estimate the system state variables asymptotically, in the presence of faults. For wind turbines, in [15], an observer-based fault detection and isolation FDI approach using a Kalman filter is developed for benchmark model. The same approach is used in [16,17] to detect sensor and actuator faults and to estimate blade root moment sensor faults and pitch actuator stuck faults, respectively. In [18], an observer-based fault detection approach for stator inter-turn short circuit fault in doubly fed electric machine, based on $dq$-frame, is established.

This paper proposes an observer-based approach for a TWT under asymmetric fault. Indeed, an active fault-tolerant control scheme based on *abc*-frame is developed in order to compensate the imbalance caused by the fault and to be stable around the nominal equilibrium. As a main feature of this method, the equipment may be protected, and subsequently the TWT continues operating under nominal conditions with acceptable performances.

The paper is organized as follows. In Section 2, the TWT in both healthy and faulty cases is described. Section 3 presents our method of active fault-tolerant control. In Section 4, simulation results highlight the interest of the proposed method. Conclusion and open problems for future research are drawn in Section 5.

## 2. Problem Statement

The model of the TWT, patented by [1], is presented on this section in both healthy and faulty cases. The principal notations used in this paper are given by Table 1.

**Table 1.** Principal notations.

| Name | Description | Name | Description |
|------|-------------|------|-------------|
| $W_w$ | wind speed | $L^h$ | inductance matrix in healthy case |
| $R_p$ | blade radius | $L^f$ | inductance matrix in faulty case |
| $\rho$ | air density | $E_{mi}$ | electromagnetic force |
| $\beta$ | pitch blade | $\Omega_i^{ref}$ | reference of the i-th angular speed |
| $\lambda$ | tip speed ratio | $\delta$ | severity of the fault |
| $r_s$ | stator resistance | $d$ | fault vector |
| $\mathcal{L}_f h$ | is the Lie derivative of $h$ with respect to $f$ | $[f,g]$ | $= \mathcal{L}_f g - \mathcal{L}_g f$ is the Lie bracket of $f$ and $g$ |

ref is used for references.

### 2.1. Healthy Twin Wind Turbine Description

As shown in Figure 1, the TWT is composed of two turbines mounted on the same tower. The advantage of this structure is that its orientation in front of the wind is ensured without needing an actuator. In fact, to keep this orientation while maintaining optimal power production requires robust control. More details on the wind turbine description can be found in [1]. The innovative concept, which is intended to be used offshore, is exposed to variable wind speeds. The mechanical power computed from the kinetic energy and the angular speed and aerodynamic torque applied to the rotor by the wind are, respectively, given by Equations (1)–(3) :

$$P_i = \frac{\pi \rho}{2} C_{pi}(\lambda_i, \beta_i) R_p^2 (W_w \cos(\psi - \alpha))^3, \tag{1}$$

$$\Omega_i = \frac{\lambda_i W_w \cos(\psi - \alpha)}{R_p}, \tag{2}$$

$$\Gamma_{a_i} = \frac{P_i}{\Omega_i} = \frac{\pi \rho}{2 \lambda_i} C_{pi}(\lambda_i, \beta_i) R_p^3 (W_w \cos(\psi - \alpha))^2, \tag{3}$$

**Remark 1.** *In this paper, the index, $i = \{1, 2\}$, denotes the i-th turbine (i.e, the first and second turbines, respectively).*

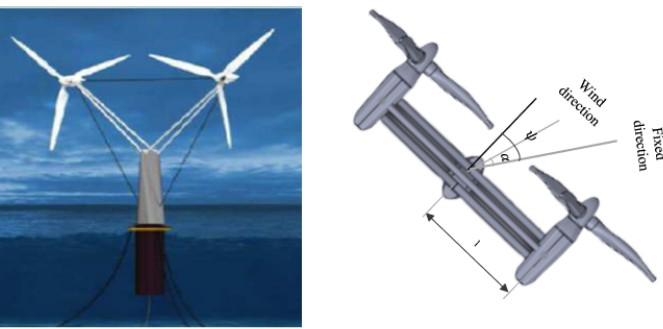

**Figure 1.** SEREO concept—face (**left**) and top (**right**) views.

The angle $\alpha$ represents the angle between the wind direction and the fixed direction and the angle $\psi$ is defined as the angle between the horizontal axis of structure and the fixed direction. $C_{pi}$ is the power coefficient [19], characterized by nonlinear functions depend on the pitch angle of blade $\beta_i$ and the tip speed ratio $\lambda_i$, which is proportional to the rotor angular speed (see Equation (2)).

To extract the maximum of wind energy, the TWT must be oriented face to wind and this is realized when the angle of orientation, $\psi$, attains its reference, $\alpha$. It can be argued that this orientation is the common input of the two turbines.

As it was mentioned, the orientation of the system in front of the wind is obtained without need of an actuator. In contrast, it is made by the torque extracted from the difference between forces generated by the two turbines, which enables system rotation around its vertical axis. Thus, considering that $\dot{\psi}$ is the derivative of $\psi$ with respect to time, dynamics of system rotation is defined by:

$$d_r \ddot{\psi} = -f_r \dot{\psi} + (F_1 - F_2)\ell, \tag{4}$$

In Equation (4), $d_r$ is the inertia moment, $f_r$ is the friction coefficient associated with yaw motion, $\ell$ is the distance between the horizontal and vertical axis of structure, and $F_1 - F_2$ is the difference between the two drag forces. The expression of $F_i$ is given by the following equation:

$$F_i = \frac{\pi \rho}{2} C_{di}(\lambda_i, \beta_i) R_p^2 (W_w \cos(\psi - \alpha))^2, \tag{5}$$

where $C_{di}$ is the drag coefficient, whose expression is given in the Appendix A by (A3).

To achieve optimal power production, power coefficients have to reach their optimum $C_{pi}{}^{opt}$, which correspond to $\beta_i{}^{opt}$ and $\lambda_i{}^{opt}$ [20]. Based on healthy assumptions, the two turbines have the same power coefficient and the same optimal tip speed ratio,

$$C_{p1}\left(\lambda_1^{opt}, \beta_1^{opt}\right) = C_{p2}\left(\lambda_2^{opt}, \beta_2^{opt}\right) \text{ and } \lambda_1^{opt} = \lambda_2^{opt} = \lambda^{opt},$$

So, the same optimal pitch angle of blades $\beta_1^{opt} = \beta_2^{opt} = \beta^{opt}$ is adopted. Furthermore, in this concept, the pitch angles are adjusted well to maximize power and to allow the rotation of the system (drag force of each turbine depends on its pitch blade (1)). The dynamics of the pitch angles can be given by the first-order system:

$$\dot{\beta}_i = \frac{1}{t_\beta}(\beta_i{}^{ref} - \beta_i), \quad i = 1, 2, \tag{6}$$

where the constant $t_\beta$ represents the time constant of blades actuation system and $\beta_i^{ref}$ is the pitch blade reference

$$\beta_1^{ref} = \beta_{opt} + \Delta\beta \text{ and } \beta_2^{ref} = \beta_{opt} - \Delta\beta.$$

The electrical part of the TWT, is equipped with two salient permanent magnet synchronous machines. This type is widely used in such applications because of many features as its high efficiency and compactness, simplification of construction and maintenance and excellent dynamic response characteristics. For the healthy case, the three-phase permanent magnet synchronous machine model in *abc*-frame [21] for each turbine, is represented by the following equations:

$$V_i = \Re_{L_i}^h I_i + L_i^h \frac{d}{dt} I_i + \frac{d}{dt} E_{mi}^h, \quad i = 1, 2, \tag{7}$$

where $\Re_{L_i}^h$ is given by:

$$\Re_{L_i}^h = \begin{bmatrix} r_s + \frac{dL_a(\theta_{ei})}{dt} & \frac{dM_{ab}(\theta_{ei})}{dt} & \frac{dM_{ac}(\theta_{ei})}{dt} \\ \frac{dM_{ba}(\theta_{ei})}{dt} & r_s + \frac{dL_b(\theta_{ei})}{dt} & \frac{dM_{bc}(\theta_{ei})}{dt} \\ \frac{dM_{ca}(\theta_{ei})}{dt} & \frac{dM_{cb}(\theta_{ei})}{dt} & r_s + \frac{dL_c(\theta_{ei})}{dt} \end{bmatrix}. \tag{8}$$

$I_i = \begin{bmatrix} i_{a_i} & i_{b_i} & i_{c_i} \end{bmatrix}^T$ and $V_i = \begin{bmatrix} v_{an_i} & v_{bn_i} & v_{cn_i} \end{bmatrix}^T$ are, respectively, the current and voltage vectors of the three phases related to the *i*-th turbine.

**Assumption 1.** *For safety reasons, the three-phase voltages are limited by $V_{max}$, for some $V_{max} > 0$.*

The electromotive force vector, $E_{mi}^h = \begin{bmatrix} e_{ma_i} & e_{mb_i} & e_{mc_i} \end{bmatrix}^T$, depends on the electrical angular position, $\theta_{ei}$ ($\frac{d\theta_{ei}}{dt} = \Omega_i$), and on the flux linkage, $\phi$, produced by the permanent magnet, where:

$$e_{ma_i} = \phi\cos(\theta_{ei}), \quad e_{mb_i} = \phi\cos\left(\theta_{ei} - \frac{2\pi}{3}\right), \quad e_{mc_i} = \phi\cos\left(\theta_{ei} + \frac{2\pi}{3}\right), \tag{9}$$

The inductance matrix $L_i^h$ also depends on the rotor electrical angular position, due to saliency:

$$L_i^h = \begin{bmatrix} L_a(\theta_{ei}) & M_{ab}(\theta_{ei}) & M_{ac}(\theta_{ei}) \\ M_{ba}(\theta_{ei}) & L_b(\theta_{ei}) & M_{bc}(\theta_{ei}) \\ M_{ca}(\theta_{ei}) & M_{cb}(\theta_{ei}) & L_c(\theta_{ei}) \end{bmatrix}, \tag{10}$$

The inductance matrix, $L_i^h$, contains self inductance of each phase in its diagonal $L_a(\theta_{ei})$, $L_b(\theta_{ei})$ and $L_c(\theta_{ei})$, and the mutual inductance, $M_{ab}(\theta_{ei})$, $M_{ac}(\theta_{ei})$ and $M_{bc}(\theta_{ei})$. Details of inductance matrix terms are carried out by the following expressions:

$$L_a(\theta_{ei}) = L_\ell + L_0 + L_1 \cos(2\theta_{ei}), \qquad M_{bc}(\theta_{ei}) = M_{cb}(\theta_{ei}) = M_0 + L_1 \cos(2\theta_{ei}), \tag{11}$$

$$L_b(\theta_{ei}), = L_\ell + L_0 + L_1 \cos(2\theta_{ei} - \frac{2\pi}{3}), \qquad M_{ab}(\theta_{ei}) = M_{ba}(\theta_{ei}) = M_0 + L_1 \cos(2\theta_{ei} - \frac{2\pi}{3}), \tag{12}$$

$$L_c(\theta_{ei}) = L_\ell + L_0 + L_1 \cos(2\theta_{ei} + \frac{2\pi}{3}), \qquad M_{ac}(\theta_{ei}) = M_{ca}(\theta_{ei}) = M_0 + L_1 \cos(2\theta_{ei} + \frac{2\pi}{3}). \tag{13}$$

where $L_\ell$, $L_0$, $M_0$, and $L_1$ are constants denoting, stator leakage inductance, average value of the winding magnetizing inductance, and the magnitude of the inductance variation due to the non-uniformity of the air gap, respectively.

**Remark 2.** *The inductance matrix is uniformly bounded over time (see Equations (11)–(13)). In addition, it admits three eigenvalues which are different from zero and not dependent on $\theta_e$. Otherwise, the determinant is a non-zero constant and then the matrix is even invertible.*

Considering that $i_{di}, i_{qi} v_{di}, v_{qi}$ are, respectively, the currents and voltages of the stator for each turbine, $L_d$ and $L_q$ are the dq-axis inductance and $p$ is the pole-pair number, in dq-frame, we have:

$$\frac{di_{di}}{dt} = -\frac{r_s}{L_d} i_{di} + \frac{pL_q}{L_d} \Omega_i i_{qi} + \frac{1}{L_d} v_{di}, \tag{14}$$

$$\frac{di_{qi}}{dt} = -\frac{r_s}{L_q} i_{qi} - \frac{pL_d}{L_q} \Omega_i i_{di} - \frac{p\phi}{L_q} \Omega_i + \frac{1}{L_q} v_{qi}. \tag{15}$$

The interconnection between the input torque (aerodynamic torque) and the output (electromagnetic torque) is ensured by the shaft of the electrical machine, and is given by the following mechanical model:

$$\frac{d\Omega_i}{dt} = \frac{1}{J}(\Gamma_{a_i} - \Gamma_{em_i} - f_v \Omega_i), \tag{16}$$

where $J$ is the total inertia, $f_v$ is the viscous friction and $\Gamma_{em_i}$ is the electromagnetic torque, given by:

$$\Gamma_{em_i} = p(L_d - L_q) i_{di} i_{qi} + p\phi i_{qi}. \tag{17}$$

Equations (14), (16) and (17) are used later in the health-control design. As previously explained, the rotation of the TWT is ensured without need of an actuator. Consequently, a control strategy must be developed in order to achieve three important objectives. To produce the maximum of electrical production, the first objective is to keep the whole structure in front of the wind to extract the maximum of energy. Secondly, the rotational speeds of the two wind turbines are controlled at a reference $\Omega_i^{ref}$, by keeping their tip–speed ratios at their optimal values. Furthermore, finally, as the fatigues loads in the mechanical shaft caused by the ripples of the electromagnetic torque can impact the produced electrical power, one solution consists of forcing the direct current $i_{d_i}$ to zero. To sum up, objectives on the healthy case can be reaching by forcing the output, $y$, to zero, where $Y$ is given by:

$$Y = \begin{bmatrix} \psi - \alpha & i_{d_1} & \Omega_1 - \Omega_{ref} & i_{d_2} & \Omega_2 - \Omega_{ref} \end{bmatrix}^T. \tag{18}$$

*2.2. Faulty Twin Wind Turbine Description*

Taking into account electrical fault, inter-turn short circuit failure in the *a*-phase has been studied in [20] to test the robustness of the control, based on the *dq*-frame. The same

fault has also been examined with a different severity in [22] to develop an active fault-tolerant control, based on the *abc*-frame.

In this paper, we assume that this fault persists on the three stator phases in only one turbine. For the healthy case, where the three stator phases are symmetric and have the same total number of winding $N_t$, the severity can be defined as the factor of the short-circuited turns $N_{a_d}$, $N_{b_d}$ and $N_{c_d}$ of the *abc*-phases devising by the total number of winding. With respect to the severities, $\delta_a$, $\delta_b$ and $\delta_c$ of *a*-phase, *b*-phase and *c*-phase, respectively, the model can be written in the *abc*-frame as:

$$V_i = \Re^f_{L_i} I_i + L^f_i \frac{d}{dt} I_i + \frac{d}{dt} E^f_{mi},$$ (19)

where the matrix $\Re^f_{L_i}$ is given by:

$$\Re^f_{L_i} = \begin{bmatrix} (1-\delta_a)\left(r_s + \frac{dL_a(\theta_{ei})}{dt}\right) & (1-(\delta_a+\delta_b))\frac{dM_{ab}(\theta_{ei})}{dt} & (1-(\delta_a+\delta_c))\frac{dM_{ac}(\theta_{ei})}{dt} \\ (1-(\delta_b+\delta_a))\frac{dM_{ba}(\theta_{ei})}{dt} & (1-\delta_b)\left(r_s + \frac{dL_b(\theta_{ei})}{dt}\right) & (1-(\delta_b+\delta_c))\frac{dM_{bc}(\theta_{ei})}{dt} \\ (1-(\delta_c+\delta_a))\frac{dM_{ca}(\theta_{ei})}{dt} & (1-(\delta_c+\delta_b))\frac{dM_{cb}(\theta_{ei})}{dt} & (1-\delta_c)\left(r_s + \frac{dL_c(\theta_{ei})}{dt}\right) \end{bmatrix},$$ (20)

The electromagnetic force vector is given by

$$E^f_{mi} = \begin{bmatrix} (1-\delta_a)e_{ma_i} & (1-\delta_b)e_{mb_i} & (1-\delta_c)e_{mc_i} \end{bmatrix}^T,$$

and the faulty inductance matrix $L^f_i$ is given by:

$$L^f_i = L^h_i - \delta_a \begin{bmatrix} L_a(\theta_{ei}) & M_{ab}(\theta_{ei}) & M_{ac}(\theta_{ei}) \\ M_{ba}(\theta_{ei}) & 0 & 0 \\ M_{ca}(\theta_{ei}) & 0 & 0 \end{bmatrix} - \delta_b \begin{bmatrix} 0 & M_{ab}(\theta_{ei}) & 0 \\ M_{ba}(\theta_{ei}) & L_b(\theta_{ei}) & M_{bc}(\theta_{ei}) \\ 0 & M_{cb}(\theta_{ei}) & 0 \end{bmatrix}$$

$$- \delta_c \begin{bmatrix} 0 & 0 & M_{ac}(\theta_{ei}) \\ 0 & 0 & M_{bc}(\theta_{ei}) \\ M_{ca}(\theta_{ei}) & M_{cb}(\theta_{ei}) & L_c(\theta_{ei}) \end{bmatrix}.$$ (21)

**Remark 3.** *For a healthy machine, the inductance has the same expression as the fault inductance just by replacing the three severities of faults with zero $\delta_a = 0$, $\delta_b = 0$, $\delta_c = 0$.*

**Assumption 2.** *The number of shorted turns is smaller than 50% of the total number of turns per phase. This implies that $\delta_a, \delta_b$ and $\delta_c \in [0, 0.5[$.*

As the fault cannot exceed the 50% (if not, then the faulty electrical machine will be drastically damaged and thus the whole system will be influenced by the defect), the time-varying inductance matrix $L^f_i$ is always invertible. In the considered case, the inverse of the matrix is carried out numerically by reason of complicated equations.

### 2.3. Control Strategy

Concerning control strategies, one is used in [23], in the healthy case, based on the high order sliding mode (HOSM) and modified in [20], when considering the faulty case (an insulation degradation fault persists on only one turbine), based on a passive fault-tolerant control. The limit of this latter control is that it can be exploited only for incipient faults. Thus, if the defect exceeds 8%, the objectives of the control are not achieved.

Nevertheless, the appearance of the fault break reflects not only the symmetry of the phases but also the symmetry of the two turbines. Consequently, if the severity of the fault is important, the stability of the TWT is disturbed, which highlights the requirement of an efficient fault detection method. Hence, an active fault-tolerant control is developed.

The idea is once the fault exceeds 8%, the system switches to the proposed observer-based active fault-tolerant control method. This latter will be able to pinpoint the inter turn short-circuit fault quickly and precisely. Concerning control strategies, one is used in [23], in the healthy case, based on high order sliding mode and modified in [20], when considering the faulty case (an insulation degradation fault persists on only one turbine), based on a passive fault-tolerant control. The limit of this latter control is that it can be exploited only for incipient faults. Thus, if the defect exceeds 8%, then the objectives of the control are not achieved.

Nevertheless, the appearance of the fault breaks not only the symmetry of phases but also the symmetry of the two turbines. Consequently, if the severity of the fault is important, the stability of the TWT is disturbed, which highlights the requirement of an efficient fault detection method. Hence, an active fault-tolerant control is developed.

The idea is once the fault exceeds 8%, the system switches to the proposed observer-based active fault-tolerant control method. This latter will be able to pinpoint the inter-turn short circuit fault quickly and precisely.

### 3. Active Fault-Tolerant Control and Diagnosis Method

*3.1. Active Fault-Tolerant Control*

In this section, the active fault-tolerant control, "when the fault is known", is developed. Considering all faults equal to zero, the proposed control is also valid for the healthy system. Taking into consideration an inter-turn short circuit on the three phases of PMSM in only one turbine, the control based on $dq$-rotating frame will not be adequate. So, a new control strategy based on $abc$-frame is adapted by depending on the direct current—indeed the homopolar component ($i_{hi} = \frac{1}{3}(i_{a_i} + i_{b_i} + i_{c_i})$)—which must be assigned to zero in order to respect the Kirchhoff laws. This approach leads to two additional states in the state vector ($i_{a_1}, i_{b_1}, i_{c_1}, i_{a_2}, i_{b_2}, i_{c_2}$ three-phase currents on each turbine are considered instead of two-phase currents $i_{d_1}, i_{q_1}, i_{d_2}, i_{q_2}$).

In the same way, two extra inputs are introduced ($v_{an_1}, v_{bn_1}, v_{cn_1}, v_{an_2}, v_{bn_2}, v_{cn_2}$ instead of $v_{d_1}, v_{q_1}, v_{d_2}, v_{q_2}$). Therefore, an extended nonlinear system based on $abc$-frame is given by:

$$\dot{x} = f(x, d) + g(x, d)u, \tag{22}$$

where expressions of $f(x, d)$ and $g(x, d)$ are given in the Appendix A (Equations (A1) and (A2)), the state vector $x$ and the input vector $u$ are given by:

$$x = \begin{bmatrix} \beta_1 & \beta_2 & \psi & \dot{\psi} & i_{a1} & i_{b1} & i_{c1} & \Omega_1 & i_{a2} & i_{b2} & i_{c2} & \Omega_2 \end{bmatrix}^T, \tag{23}$$

$$u = \begin{bmatrix} \Delta\beta & v_{an_1} & v_{bn_1} & v_{cn_1} & v_{an_2} & v_{bn_2} & v_{cn_2} \end{bmatrix}^T. \tag{24}$$

**Remark 4.** *As $\Delta\beta$ represents the difference between the two blade pitch angles, which is limited by $2\pi$, and the three-phase voltage, $V_i$, which is also limited by $V_{max}$, the input vector u in our case is bounded.*

Comparing to [22], the active fault-tolerant control has 7 objectives to achieve instead of 5. In fact, the two additional outputs $i_{h1} = \frac{1}{3}(x_5 + x_6 + x_7)$ and $i_{h2} = \frac{1}{3}(x_9 + x_{10} + x_{11})$ are used to correct the asymmetry of phases and thus the expression of the new outputs vector:

$$y = h(x) = \begin{bmatrix} x_3 - \alpha & i_{d1} & x_8 - \Omega_1^{ref} & i_{h1} & i_{d2} & x_{12} - \Omega_2^{ref} & i_{h2} \end{bmatrix}^T, \tag{25}$$

Note that $i_{d1}$ and $i_{d2}$ depend, respectively, on ($x_5, x_6, x_7$) and ($x_9, x_{10}, x_{11}$), with respect to Park's transformation. The successive derivatives of the outputs lead to the following equation:

$$y^{(\varepsilon)} = \Lambda(x, d) + \Theta(x, d)u(x, d), \tag{26}$$

where $y^{(\varepsilon)} = (y_1^{(3)}, y_2^{(1)}, y_3^{(2)}, y_4^{(1)}, y_5^{(1)}, y_6^{(2)}, y_7^{(1)})^T$, with $y^{(i)} = \frac{d^i y}{dt^i}$, for $i \geq 1$. The control is developed based on high order sliding mode, for that each output has its relative degree $\varepsilon = (\varepsilon_1, \varepsilon_2, \varepsilon_3, \varepsilon_4, \varepsilon_5, \varepsilon_6, \varepsilon_7) = (3, 1, 2, 1, 1, 2, 1)$ related to $(y_1, y_2, y_3, y_4, y_5, y_6, y_7)$, respectively.

The vector field $\Lambda$ is given in the Appendix A (A5) and the decoupling matrix $\Theta$ is given by:

$$\Theta(x, d) = \begin{bmatrix} \Theta_1 & & 0 \\ & \Theta_2 & \\ 0 & & \Theta_3 \end{bmatrix} \qquad \begin{cases} \Theta_1 = \frac{-2\rho\pi(W_w \cos(x_3 - \alpha))^2}{d_r T_\beta} B(\lambda) \\ \Theta_2 = P_i L_i^{f^{-1}} \\ \Theta_3 = P_i L_i^{h^{-1}}, \end{cases} \tag{27}$$

where $P_i$ is the Park's transformation matrix. The decoupling matrix, $\Theta$, is a square block diagonal matrix in which the diagonal elements are three square matrices of $(1 \times 1, 3 \times 3$ and $3 \times 3)$ and the off-diagonal elements are 0. The second block is the product of the Park transformation matrix and the inverse of inductance matrix of the faulty turbine $L_i^f$ and the third block is the product of the Park transformation matrix and the inverse of inductance matrix of the healthy one $L_i^h$. As a result, this matrix is regular, which leads to a following decoupling control based on a new control $\vartheta$ (which will be detailed later):

$$u(x, d) = \Theta(x, d)^{-1}(\vartheta - \Lambda(x, d)). \tag{28}$$

**Diffeomorphism**

From the previously presented control scheme, we define the diffeomorphism $Z = h(x)$, which given as follows $z_1 = y_1, z_2 = \dot{y}_1, z_3 = \ddot{y}_1, z_4 = y_2, z_5 = y_3, z_6 = \ddot{y}_3, z_7 = y_4, z_8 = y_5$, $z_9 = y_6, z_{10} = \dot{y}_6, z_{11} = y_7$ and $z_{12} = \beta_1 + \beta_2$.

$$\begin{cases} z_1 = x_3 - \alpha \\ z_2 = x_4 \\ z_3 = -\frac{f_r}{d_r} x_4 + \frac{2\pi l \rho R_p^2}{d_r} B(\lambda) W_w^2 (\cos(x_3 - \alpha))^2 (x_1 - x_2) \\ z_4 = \sqrt{\frac{2}{3}} \left( \cos(\theta_e) x_5 + \cos(\theta_e - \frac{2\pi}{3}) x_6 + \cos(\theta_e + \frac{2\pi}{3}) x_7 \right) \\ z_5 = x_8 - \Omega_1^{ref} \\ z_6 = \frac{1}{J}(\Gamma_{a1} - \Gamma_{em1} - f_v x_8) \\ z_7 = \sqrt{\frac{1}{3}}(x_5 + x_6 + x_7) \\ z_8 = \sqrt{\frac{2}{3}} \left( \cos(\theta_e) x_9 + \cos(\theta_e - \frac{2\pi}{3}) x_{10} + \cos(\theta_e + \frac{2\pi}{3}) x_{11} \right) \\ z_9 = x_{12} - \Omega_2^{ref} \\ z_{10} = z_6 = \frac{1}{J}(\Gamma_{a1} - \Gamma_{em1} - f_v x_{12}) \\ z_{11} = \sqrt{\frac{1}{3}}(x_9 + x_{10} + x_{11}) \\ z_{12} = x_1 + x_2 \end{cases} \tag{29}$$

Knowing that $\sum\limits_{k=1}^{7} \varepsilon_k = 11 < 12$, there is a zero dynamics of dimension 1, which is expressed as:

$$\dot{z}_{12} = \frac{2\beta^{ref}}{t_\beta} - \frac{1}{t_\beta} z_{12}. \tag{30}$$

The zero dynamics depends on the external input which is not directly influenced by the defect. For that, the dynamic of zero, $z_{12}$, is always input-to-state stable (input: $\beta^{ref}$) [24].

**Definition 1.** *Consider an integer, $\epsilon \geq 2$, and for $j = 2, \ldots \epsilon$, define the vector, $Z_j = [z_1, \ldots z_j]^T$, the decreasing sequence of positive real numbers, $r_j = r_1 - (j - 1)$, for some $r_1 \geq 2$, the non*

*decreasing sequence of positive real numbers $\alpha_j$, so that $\alpha_j \geq \alpha_{j-1} \geq \cdots \geq r_1$. Defining also, for $j = 2, \ldots \epsilon$, the $C^1$ r-homogeneous function:*

$$\sigma_j = \lceil z_j \rfloor^{\frac{\alpha_j}{r_j}} + k_{j-1}^{\frac{\alpha_j}{r_j}} \lceil \sigma_{j-1} \rfloor^{\frac{\alpha_j}{\alpha_{j-1}}} \tag{31}$$

*where $\lceil \rfloor^c$ being a function defined as $|.|^c.sgn(.)$, the index $\sigma_1 = \lceil z_1 \rfloor^{\frac{\alpha_1}{r_1}}$ and $k_j \geq 0$.*

Based on the definition and the diffeomorphism, the proposed controller in this paper reads as:

$$\vartheta = -K_y \lceil \sigma_y \rfloor^{\mu_{j,k}} = \begin{bmatrix} -K_\psi \lceil \sigma_\psi \rfloor^{\mu_{1,3}} \\ -K_{i_d} \lceil z_4 \rfloor^{\mu_4} \\ -K_\Omega \lceil \sigma_{\Omega_1} \rfloor^{\mu_{5,6}} \\ -K_{i_h} \lceil z_7 \rfloor^{\mu_7} \\ -K_{i_d} \lceil z_8 \rfloor^{\mu_8} \\ -K_\Omega \lceil \sigma_{\Omega_2} \rfloor^{\mu_{9,10}} \\ -K_{i_h} \lceil z_{11} \rfloor^{\mu_{11}} \end{bmatrix}, \tag{32}$$

where $\sigma_p si$, $\sigma_{\Omega_1}$, $\sigma_{\Omega_2}$ can be calculated based on Definition 1 and the parameter $\mu$ reads as the following adaptive law:

$$\mu_{j,k} = \max\{1 - \gamma \sum_j^k \frac{|z_j|}{|z_j| + c}, 0\}. \qquad 1 \leq j, k \leq 11, \qquad c > 0, \gamma > 1$$

The advantage of the active fault-tolerant control, that it makes the machine behave as in the case where the three phases are symmetric and that way it enables the fault to be compensated. In this section, we assume that the fault is known. Because this is not a realistic case, we will treat the global problem in the next sections.

### 3.2. Fault Estimation

In this section, a fault detection method is developed in order to estimate the defect, *d*. Coupled with the control designed in Section 3.2, the fault detection method gives the possibility to design an active fault-tolerant control in order to achieve the reference trajectory. The global fault vector, *d*, that contains the three-phase defects is carried out by:

$$d = \begin{bmatrix} \delta_a & \delta_b & \delta_c \end{bmatrix}^T. \tag{33}$$

The monitoring sensors in the faulty model are $x_5$, $x_6$ and $x_7$ (which correspond, respectively, to the three-phase currents $i_a$, $i_b$ and $i_c$).

The idea here is to reorganize terms which are expressed linearly in function of the defect in the right hand and which are independent on the fault in the left. Otherwise, from Equations (7) and (19), and knowing that the defect is modeled in a linear way under the following form:

$$\begin{cases} \Re_{L_i}^f = \Re_{L_i}^h - \Delta\Re_L; \quad \Delta\Re_L = \begin{bmatrix} \delta_a\left(r_s + \frac{dL_a(\theta_{ei})}{dt}\right) & (\delta_a + \delta_b)\frac{dM_{ab}(\theta_{ei})}{dt} & (\delta_a + \delta_c)\frac{dM_{ac}(\theta_{ei})}{dt} \\ (\delta_b + \delta_a)\frac{dM_{ba}(\theta_{ei})}{dt} & \delta_b\left(r_s + \frac{dL_b(\theta_{ei})}{dt}\right) & (\delta_b + \delta_c)\frac{dM_{bc}(\theta_{ei})}{dt} \\ (\delta_c + \delta_a)\frac{dM_{ca}(\theta_{ei})}{dt} & (\delta_c + \delta_b)\frac{dM_{cb}(\theta_{ei})}{dt} & \delta_c\left(r_s + \frac{dL_c(\theta_{ei})}{dt}\right) \end{bmatrix} \\[3em] L_i^f = L_i^h - \Delta L; \quad \Delta L = \begin{bmatrix} \delta_a L_a(\theta_{ei}) & (\delta_a + \delta_b)M_{ab}(\theta_{ei}) & (\delta_a + \delta_c)M_{ac}(\theta_{ei}) \\ (\delta_b + \delta_a)M_{ba}(\theta_{ei}) & \delta_b L_b(\theta_{ei}) & (\delta_b + \delta_c)M_{bc}(\theta_{ei}) \\ (\delta_c + \delta_a)M_{ca}(\theta_{ei}) & (\delta_c + \delta_b)M_{cb}(\theta_{ei}) & \delta_c L_c(\theta_{ei}) \end{bmatrix} \\[3em] E_{mi}^f = E_{mi}^h - \Delta E_m; \quad \Delta E_m = \begin{bmatrix} \delta_a\phi\cos(\theta_{ei}) & \delta_b\phi\cos\left(\theta_{ei} - \frac{2\pi}{3}\right) & \delta_c\phi\cos\left(\theta_{ei} + \frac{2\pi}{3}\right) \end{bmatrix}^T \end{cases}$$

We can obtain relation (34) between the measurement vector, $\gamma$, and the fault vector, $d$, which takes the form $\gamma = D(x)d$.

$$\underbrace{V_i - \Re_{L_i}^h I_i + L_i^h \frac{d}{dt} I_i + \frac{d}{dt} E_{mi}^h}_{\gamma} = \underbrace{\Delta \Re_L I_i + \Delta L \frac{d}{dt} I_i + \Delta E_m}_{D(x)d}, \tag{34}$$

where D(x) is given by:

$$D(x) = \begin{bmatrix} D_{11} & R_{Li}^h(1,2)i_{bi} + L_i^h(1,2)\frac{di_{bi}}{dt} & R_{Li}^h(1,3)i_{ci} + L_i^h(1,3)\frac{di_{ci}}{dt} \\ R_{Li}^h(2,1)i_{ai} + L_i^h(2,1)\frac{di_{ai}}{dt} & D_{22} & R_{Li}^h(2,3)i_{ci} + L_i^h(2,3)\frac{di_{ci}}{dt} \\ R_{Li}^h(3,1)i_{ai} + L_i^h(3,1)\frac{di_{ai}}{dt} & R_{Li}^h(3,2)i_{bi} + L_i^h(3,2)\frac{di_{bi}}{dt} & D_{33} \end{bmatrix}, \tag{35}$$

$$\text{For} \quad D_{kk} = \sum_{j=1}^{3} R_{Li}^h(k,j)I_i(j) + L_i^h(k,j)\frac{dI_i(j)}{dt} + E_{mi}(k). for \quad k = 1,2,3 \tag{36}$$

Note that $R_{Li}^h(k,j)$ refers to the element of the row, $k$, and the column, $j$, is related to the matrix, $R_{Li}^h$.

**Remark 5.** *If the matrix $D(x)$ is regular, it is possible at the same time to solve the problem of detection, isolation and estimation of the faults.*

**Assumption 3.** *The matrix $D(x)$ is always invertible.*

It is proved, numerically (because of the equation's complexity), that $D(x)$ is invertible. In our case, the number of measurements is equal to the number of unknown uncertainties, which can solve the left inverse problem by inverting directly the matrix $D(x)$ in (34).

In this work, the estimation method is stable and the active fault-tolerant control is also stable. Even so, what can we say about the global stability in the case of the coupling of the two tools?

*3.3. Observer-Based Active Fault-Tolerant Control*

The diagram in Figure 2 provides a comprehensive summary of inputs and outputs of the system, the concept of the proposed technique and the entire system, which is made up of two turbines: a healthy turbine and a faulty turbine. After verifying the stability of both control strategy and the fault estimation method, the two strategies are coupled in order to detect the fault and to correct it at the same time.

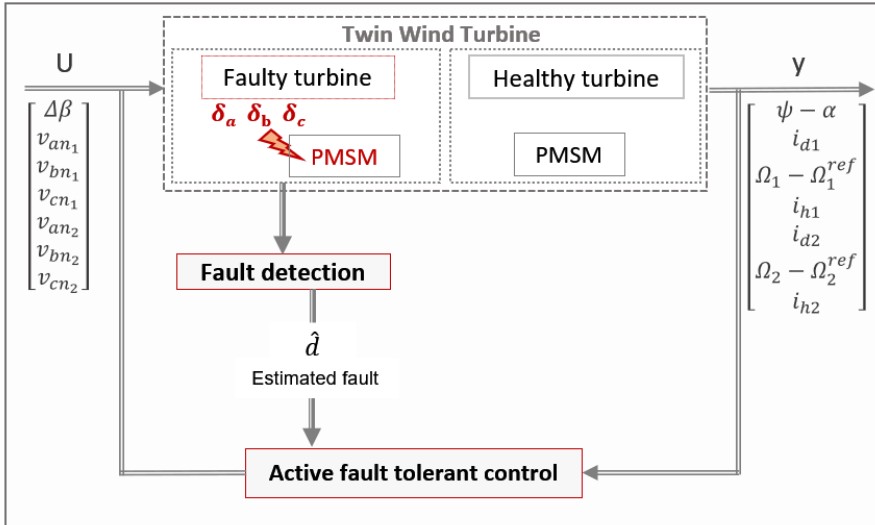

**Figure 2.** Diagnosis and fault-tolerant control diagram.

The diagnosis method focus only on one part of the global system, the dynamics of observers—indeed the global model—are carried out by the following system:

$$\begin{cases} \dot{x} = f(x,d) + g(x,d)u(x,\hat{d}) \\ \dot{\hat{d}} = -\left|d - \hat{d}\right|^{0.5} sign\left(d - \hat{d}\right) \end{cases} \tag{37}$$

The power coefficient 0.5 is used in order to reduce sensitivity to noise [25]. Neverthe-less, if it is equal to 0, then the dynamical system becomes a first order sliding mode, and the chattering phenomenon occurs at the equilibrium point. Furthermore, if this coefficient is equal to 1, then the dynamical system becomes linear.

**Remark 6.** *In* (37)*, d comes from the algebraic resolution of* (34)*, if we consider that this resolution is carried out without computation and measurement noise, then we can use directly d in the active fault-tolerant control, and there is no more stability problem due to the fact that $\hat{d}$ is not always equal to d in the control.*

**Theorem 1.** *Under Assumptions* 1*, A2 and A3, the closed-loop system* (37) *is exponentially stable.*

**Proof of Theorem 1.** Note that, since the zero dynamic (i.e., (30)) is only exponentially stable and does not depend on the fault $d$, the stability is proved only exponentially even if all the other states are finite time stable.

Now, considering $\hat{d} = d$, the first equation of the system (37) becomes:

$$\dot{x} = f(x,d) + g(x,d)u(x,d), \tag{38}$$

and this gives $y^{(\varepsilon)} = \vartheta$, which is finitely time stable, see [26]. Moreover, the dynamic $\dot{\hat{d}} = -\left|d - \hat{d}\right|^{0.5} sign\left(d - \hat{d}\right)$ is also finitely time stable.

Consequently, it is only sufficient to verify that during the time $\hat{d}$ is different to $d$, the state $x$ remains bounded. This is equivalent to prove that the system (39) is locally ISS with respect to $\Delta u$.

$$\dot{x} = f(x,d) + g(x,d)u(x,\hat{d}) = f(x,d) + g(x,d)u(x,d) + g(x,d)\Delta u. \tag{39}$$

where $\Delta u = u(x,\hat{d}) - u(x,d)$.

Note that $\Delta u$ is bounded by Assumption 1 and $g(x,d)$ is bounded under the Assumption 2 (which is also detailed in the Appendix A). The local ISS property of the previous equation

follows from the fact that, when $\hat{d} = d$ ($\Delta u = 0$), the system (38) (or equivalently (29) in Z) is finitely time stable (and therefore at least locally exponentially stable) together with the fact that $g(x, d)\Delta u$ is bounded (see [27] for more details).

This ends the proof of the theorem. $\square$

## 4. Simulation Results

The faulty TWT model extracted in the problem statement has been applied to evaluate the performances of the proposed diagnosis method. In this section, simulation results, using Ode1 solver in Matlab Simulink (Mathworks Inc., Natick, MA, USA) with a fixed time step of $10^{-4}$ s, are performed and commented.

To quantify the influence of the faults on the outputs of the system, an inter-turn short circuit fault is considered on one phase with a severity of 4%, which occurs after 7 s. A passive control based on HOSM is used in this case [20]. As can be seen in Figure 3a, once the fault appears, the three-phase currents of the faulty machine increase and the direct current is not well controlled to be assigned to zero (Figure 3b). This is because the fault breaks the symmetry of the three phases of the machine. In Figure 4, the angular speed of the healthy machine follows its reference (the blue and the green curves are confused), whereas the angular speed of the faulty machine does not reach its reference. Therefore, if the fault exceeds 8%, then a new control method is indispensable to compensate the fault and to keep the TWT in front of the wind. For that, an active fault-tolerant control is proposed.

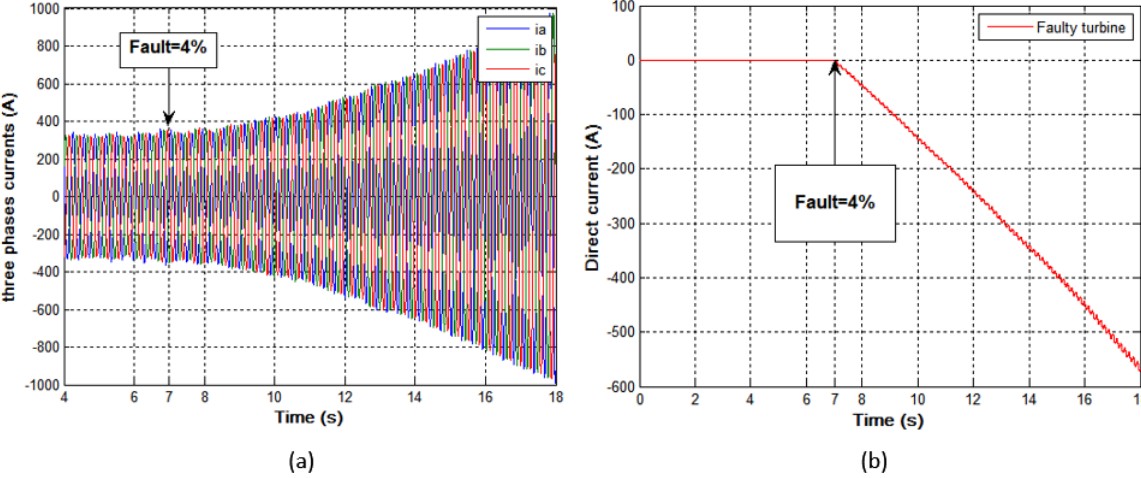

(a)                                                                                        (b)

**Figure 3.** The effect of an inter-turn short circuit fault of severity 4% at time 7 s before the AFTC: (**a**) the three-phase currents of the faulty machine and (**b**) the direct current of the faulty machine.

To prove the effectiveness of the proposed method, three faults are considered, respectively, in the *a*-phase, *b*-phase and *c*-phase of the PMSM of only one turbine, with important severities which are greater than 8% (30%, 16% and 10% of turns are short-circuited, respectively, in the *a*-phase, *b*-phase and *c*-phase).

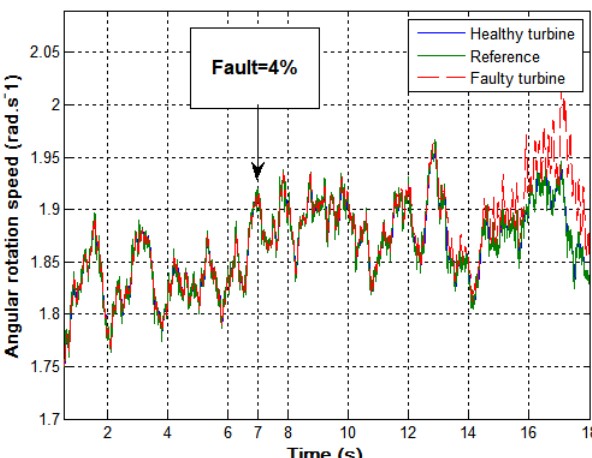

**Figure 4.** Comparison between angular speeds of the healthy and the faulty machines before the AFTC.

From Figure 5, the first fault is activated at the time 3 s, which affects the *a*-phase, while the other faults are set to zero.

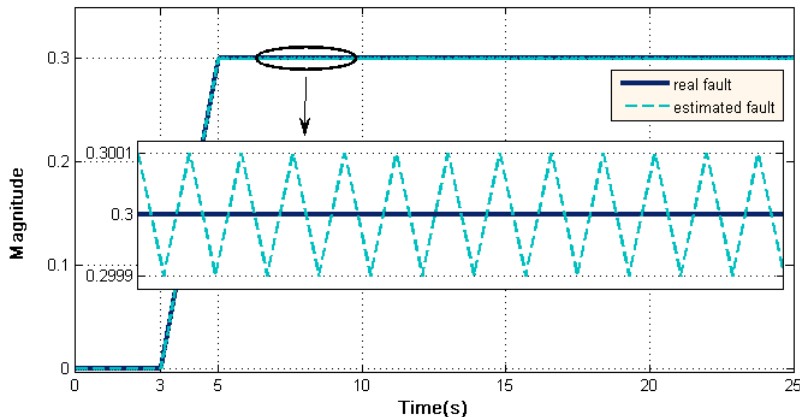

**Figure 5.** Diagnosis of inter-turn short circuit fault in the electrical machine of TWT: 30% of *a*-phase turns are defected at $t = 3$ s.

After 3 s, 30% of the totality of turns is defected. At time 5 s, the second fault is started and then after 8 s the third fault is setting off, which leads to the existence of three faults simultaneously. For the second fault, 16% of turns are defected, whereas only 10% of turns are defected on the *c*-phase (see Figure 6).

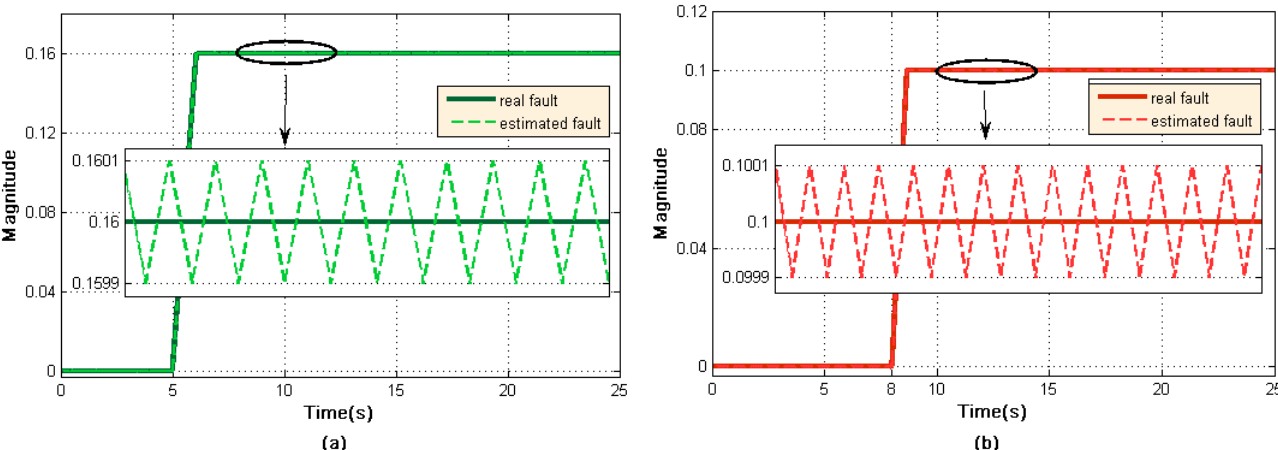

**Figure 6.** Diagnosis of inter-turn short circuit fault in the electrical machine of TWT: (**a**) severity = 16% of: *b*-phase at $t = 5$ sand (**b**) severity = 10% of: *c*-phase at $t = 8$ s.

In this case, a diagnosis method has been carried out: firstly, we are able to detect the presence of faults, secondly, to identify and distinguish them from the obtained signals and finally, the active fault-tolerant control has been implemented in order to ensure a well-functioning of the system. With the presence of faults, the three-phase currents are subject to increase and generate an uncontrolled direct current, $i_{d_i}$. With the AFTC, these currents are well guided to attain their references, as shown in Figure 7.

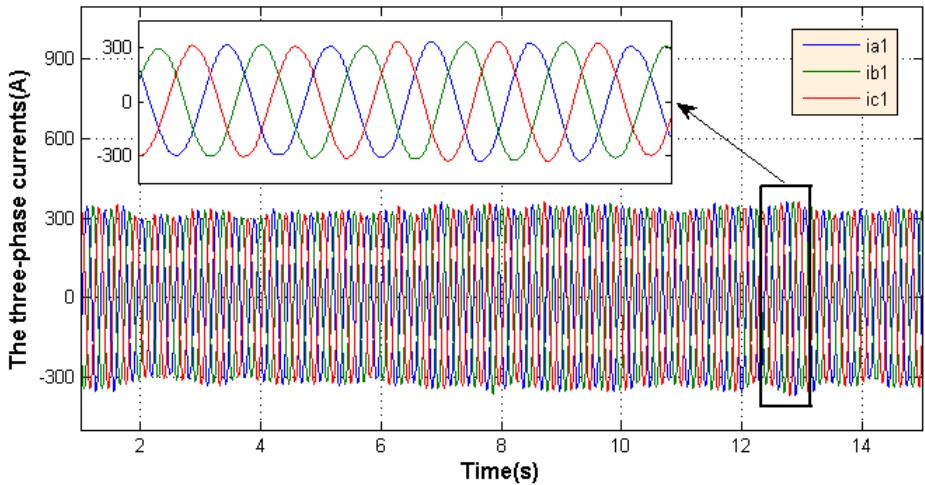

**Figure 7.** The effectiveness of the active fault-tolerant control on the three-phase currents of faulty PMSM.

## 5. Conclusions

In this paper, an active fault-tolerant control based on fault estimation was proposed for a new concept of a TWT. Numerical simulations performed in MATLAB/Simulink proved that the proposed AFTC design can ensure the stability of the electrical machine currents output with and without a fault. In addition, the proposed approach makes it possible to keep the whole system oriented to face toward the wind, despite the existence of a fault on only one turbine. Moreover, the fault detection approach detailed in this paper can be used for many other faults affecting wind turbines. For that, further work will be required to apply the sparse recovery diagnosis method, while considering other types of faults related to the operating conditions which the turbine is connected to.

**Author Contributions:** Methodology, I.H., J.-P.B. and F.P.; Supervision, I.H. and J.-P.B.; Validation, I.H. and J.-P.B.; Visualization, N.F. and M.S.A.; Writing—original draft, M.M.; Writing—review & editing, I.H. and J.-P.B. All authors have read and agreed to the published version of the manuscript.

**Funding:** This research received no external funding.

**Institutional Review Board Statement:** Not applicable.

**Informed Consent Statement:** Not applicable.

**Data Availability Statement:** Not applicable.

**Conflicts of Interest:** The authors decalre no conflict of interest.

## Abbreviations

The following abbreviations are used in this manuscript:

| | |
|---|---|
| TWT | Twin Wind Turbine |
| PMSM | Permanent Magnet Synchronous Machine |
| MCSA | Motor Current Signature Analysis |
| DWT | Discrete Wavelet Transforms |
| CWT | Continuous Wavelet Transforms |
| FDI | Fault Detection and Isolation |
| AFTC | Active Fault-Tolerant Control |
| ISS | Input to-State Stable |
| HOSM | High Order Sliding Mode |

## Appendix A

The vector $f(x,d)$ and the matrix $g(x,d)$ presented on the nonlinear system (22) are expressed as:

$$f(x,d) = \begin{bmatrix} \frac{\beta_1^{opt} - x_1}{T_\beta} \\ \frac{\beta_2^{opt} - x_2}{T_\beta} \\ x_4 \\ -\frac{1}{d_r}(f_r x_4 + (F_1(x_1, x_3) - F_2(x_2, x_3)))L \\ \left\{ L_1^{f-1}\left( \Re_{L_1}^f \begin{bmatrix} x_5 & x_6 & x_7 \end{bmatrix}^T + \frac{d}{dt} E_{m_1}^f \right) \right\} \\ \frac{1}{J}(\Gamma_{a1}(x_1, x_3) - \Gamma_{em1}(x_5, x_6, x_7) - f_v x_8) \\ \left\{ L_2^{h-1}\left( \Re_{L_2}^h \begin{bmatrix} x_9 & x_{10} & x_{11} \end{bmatrix}^T + \frac{d}{dt} E_{m_2}^h \right) \right\} \\ \frac{1}{J}(\Gamma_{a1}(x_2, x_3) - \Gamma_{em1}(x_9, x_{10}, x_{11}) - f_v x_{12}) \end{bmatrix}, \qquad (A1)$$

$$g(x,d) = \begin{bmatrix} \frac{1}{T_\beta} & 0 & 0 & 0 & 0 & 0 & 0 \\ -\frac{1}{T_\beta} & 0 & 0 & 0 & 0 & 0 & 0 \\ 0 & 0 & 0 & 0 & 0 & 0 & 0 \\ 0 & 0 & 0 & 0 & 0 & 0 & 0 \\ 0 & & & & 0 & 0 & 0 \\ 0 & & L_1^{f^{-1}} & & 0 & 0 & 0 \\ 0 & & & & 0 & 0 & 0 \\ 0 & 0 & 0 & 0 & 0 & 0 & 0 \\ 0 & 0 & 0 & 0 & & & \\ 0 & 0 & 0 & 0 & & L_2^{h^{-1}} & \\ 0 & 0 & 0 & 0 & & & \\ 0 & 0 & 0 & 0 & 0 & 0 & 0 \end{bmatrix}. \tag{A2}$$

As $T_\beta$ is a constant, the two matrix $L_1^f$ and $L_2^h$ are also bounded and always invertible. This implies that the matrix $g(x,d)$ is bounded by the inverse of their maximum $\max\left(L_1^f, L_2^h\right) = L_l + L_0 + L_1$.

The expression of the drag coefficient $c_{d_i}$ is given by:

$$c_{d_i}(\lambda_i, \beta_i) = A_i(\lambda_i) + B_i(\lambda_i)\beta_i, \tag{A3}$$

where

$$A_i(\lambda_i) = a_0 + a_1\lambda_i + a_2\lambda_i^2 + a_3\lambda_i^3 \quad \text{and} \quad B_i(\lambda_i) = b_0 + b_1\lambda_i + b_2\lambda_i^2 + b_3\lambda_i^3. \tag{A4}$$

The constants $a_k$ and $b_k$ for $k = 1, 2, 3$ are equal, respectively, to: $a_0 = 0.25382$, $a_1 = -0.1369$, $a_2 = 0.04345$, $a3 = -0.00263$, $b_0 = -0.008608$, $b_1 = 0.0063$, $b_2 = -0.0015$ and $b_3 = 0.000118$.

Considering that $C = \frac{\rho\pi}{2}R_p^2 W_w^2 \cos^2(\psi - \alpha)$, and with repect to the fault, the vector field $\Lambda(x,d)$ is given by:

$$\Lambda(x,d) = \begin{bmatrix} -\frac{f_r}{d_r}\ddot{\psi} + \frac{B(\lambda)C}{d_r T_\beta}l(\beta_1 - \beta_2) + \frac{B(\lambda)\dot{C}}{d_r}l(\beta_1 - \beta_2) + \frac{\dot{B}(\lambda)C}{d_r}l(\beta_1 - \beta_2) \\ \Lambda_2 \\ \frac{1}{J}\left(\dot{\Gamma}_{a_1} - \dot{\Gamma}_{em_1}\right) \\ \sqrt{\frac{1}{3}}(f_5 + f_6 + f_7) \\ \Lambda_5 \\ \frac{1}{J}\left(\dot{\Gamma}_{a_2} - \dot{\Gamma}_{em_2}\right) \\ \sqrt{\frac{1}{3}}(f_9 + f_{10} + f_{11}) \end{bmatrix}, \tag{A5}$$

with

$$\Lambda_2 = \sqrt{\frac{2}{3}}\left(\begin{bmatrix} \cos(\theta_{e1}) & \cos\left(\theta_{e1} - \frac{2\pi}{3}\right) & \cos\left(\theta_{e1} + \frac{2\pi}{3}\right) \end{bmatrix}\begin{bmatrix} f_5 & f_6 & f_7 \end{bmatrix}^T - px_8\begin{bmatrix} \sin(\theta_{e1}) & \sin\left(\theta_{e1} - \frac{2\pi}{3}\right) & \sin\left(\theta_{e1} + \frac{2\pi}{3}\right) \end{bmatrix}\begin{bmatrix} x_5 & x_6 & x_7 \end{bmatrix}^T\right), \tag{A6}$$

$$\Lambda_5 = \sqrt{\frac{2}{3}}\left(\begin{bmatrix} \cos(\theta_{e2}) & \cos\left(\theta_{e2} - \frac{2\pi}{3}\right) & \cos\left(\theta_{e2} + \frac{2\pi}{3}\right) \end{bmatrix}\begin{bmatrix} f_9 & f_{10} & f_{11} \end{bmatrix}^T - px_{12}\begin{bmatrix} \sin(\theta_{e2}) & \sin\left(\theta_{e2} - \frac{2\pi}{3}\right) & \sin\left(\theta_{e2} + \frac{2\pi}{3}\right) \end{bmatrix}\begin{bmatrix} x_9 & x_{10} & x_{11} \end{bmatrix}^T\right).$$

The derivative of the two electromagnetic torques are given, respectively, by these expressions:

$$\dot{\Gamma}_{em_1} = \sigma_{21}\Lambda_2 + \sigma_{22}\Lambda_{21} \quad \text{and} \quad \dot{\Gamma}_{em_2} = \sigma_{51}\Lambda_5 + \sigma_{52}\Lambda_{51}. \text{ with}$$

$$\Lambda_{21} = -\sqrt{\frac{2}{3}}\left(\begin{bmatrix} \sin(\theta_{e1}) & \sin(\theta_{e1} - \frac{2\pi}{3}) & \sin(\theta_{e1} + \frac{2\pi}{3}) \end{bmatrix}\begin{bmatrix} f_5 & f_6 & f_7 \end{bmatrix}^T - px_8\begin{bmatrix} \cos(\theta_{e1}) & \cos(\theta_{e1} - \frac{2\pi}{3}) & \cos(\theta_{e1} + \frac{2\pi}{3}) \end{bmatrix}\begin{bmatrix} x_5 & x_6 & x_7 \end{bmatrix}^T\right), \tag{A7}$$

$$\Lambda_{51} = -\sqrt{\frac{2}{3}}\left(\begin{bmatrix} \sin(\theta_{e2}) & \sin(\theta_{e2} - \frac{2\pi}{3}) & \sin(\theta_{e2} + \frac{2\pi}{3}) \end{bmatrix}\begin{bmatrix} f_9 & f_{10} & f_{11} \end{bmatrix}^T - px_{12}\begin{bmatrix} \cos(\theta_{e2}) & \cos(\theta_{e2} - \frac{2\pi}{3}) & \cos(\theta_{e2} + \frac{2\pi}{3}) \end{bmatrix}\begin{bmatrix} x_9 & x_{10} & x_{11} \end{bmatrix}^T\right), \tag{A8}$$

$$\sigma_{21} = p\phi_f + \sqrt{\frac{2}{3}}p(L_d - L_q)\begin{bmatrix} \cos(\theta_{e1}) & \cos(\theta_{e1} - \frac{2\pi}{3}) & \cos(\theta_{e1} + \frac{2\pi}{3}) \end{bmatrix}\begin{bmatrix} x_5 & x_6 & x_7 \end{bmatrix}^T, \tag{A9}$$

$$\sigma_{22} = -\sqrt{\frac{2}{3}}p(L_d - L_q)\begin{bmatrix} \sin(\theta_{e1}) & \sin(\theta_{e1} - \frac{2\pi}{3}) & \sin(\theta_{e1} + \frac{2\pi}{3}) \end{bmatrix}\begin{bmatrix} x_5 & x_6 & x_7 \end{bmatrix}^T, \tag{A10}$$

$$\sigma_{51} = p\phi_f + \sqrt{\frac{2}{3}}p(L_d - L_q)\begin{bmatrix} \cos(\theta_{e2}) & \cos(\theta_{e2} - \frac{2\pi}{3}) & \cos(\theta_{e2} + \frac{2\pi}{3}) \end{bmatrix}\begin{bmatrix} x_9 & x_{10} & x_{11} \end{bmatrix}^T, \tag{A11}$$

$$\sigma_{52} = -\sqrt{\frac{2}{3}}p(L_d - L_q)\begin{bmatrix} \sin(\theta_{e2}) & \sin(\theta_{e2} - \frac{2\pi}{3}) & \sin(\theta_{e2} + \frac{2\pi}{3}) \end{bmatrix}\begin{bmatrix} x_9 & x_{10} & x_{11} \end{bmatrix}^T. \tag{A12}$$

The park transformation matrix $P_i$ is given by:

$$P_i = \sqrt{\frac{2}{3}}\begin{bmatrix} \cos(\theta_{ei}) & \cos(\theta_{ei} - \frac{2\pi}{3}) & \cos(\theta_{ei} + \frac{2\pi}{3}) \\ -\sin(\theta_{ei}) & -\sin(\theta_{ei} - \frac{2\pi}{3}) & -\sin(\theta_{ei} + \frac{2\pi}{3}) \\ \sqrt{\frac{1}{2}} & \sqrt{\frac{1}{2}} & \sqrt{\frac{1}{2}} \end{bmatrix}. \tag{A13}$$

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
