# Peer review of "Observer-Based Active Fault-Tolerant Control of an Asymmetric Twin Wind Turbine"

_information, doi:10.3390/info13030113_

Round 1

Reviewer 1 Report

See attached report

Author Response

Dear Reviewer,

Thank  you for giving us the opportunity to revise our manuscript.  Please find attached our answers point by point. For the ease of review, the main changes are highlighted in the revised version via red color.

Best regards

Reviewer 2 Report

Comments can be seen in  attachment.

Author Response

(The authors gave the same response as above.)

Reviewer 3 Report

This paper presents the design of an active fault-tolerant control based on observers for a twin wind turbine consisting of two wind turbines. The authors consider an asymmetric conditions case when only one turbine is affected by an inter-turn short circuit fault of a permanent magnet synchronous machine. A diagnosis design is developed. The main purpose of the proposed model is to detect and correct the considered fault in a short time.

The presentation of the paper is well. The paper is Well written. The paper is accepted for publication.

Author Response

Dear Reviewer,

We appreciate the time and effort that you have dedicated, and we thank you for accepting our manuscript.

Best regards

Round 2

Reviewer 2 Report

No comment.